# Unnatural amino acid photo-crosslinking of the $I_{Ks}$ channel complex demonstrates a KCNE1:KCNQ1 stoichiometry of up to 4:4

Christopher I Murray, Maartje Westhoff, Jodene Eldstrom, Emely Thompson, Robert Emes, David Fedida*

Department of Anesthesiology, Pharmacology and Therapeutics, University of British Columbia, Vancouver, Canada

**Abstract** Cardiac repolarization is determined in part by the slow delayed rectifier current ($I_{Ks}$), through the tetrameric voltage-gated ion channel, KCNQ1, and its β-subunit, KCNE1. The stoichiometry between α and β-subunits has been controversial with studies reporting either a strict 2 KCNE1:4 KCNQ1 or a variable ratio up to 4:4. We used $I_{Ks}$ fusion proteins linking KCNE1 to one (EQ), two (EQQ) or four (EQQQQ) KCNQ1 subunits, to reproduce compulsory 4:4, 2:4 or 1:4 stoichiometries. Whole cell and single-channel recordings showed EQQ and EQQQQ to have increasingly hyperpolarized activation, reduced conductance, and shorter first latency of opening compared to EQ - all abolished by the addition of KCNE1. As well, using a UV-crosslinking unnatural amino acid in KCNE1, we found EQQQQ and EQQ crosslinking rates to be progressively slowed compared to KCNQ1, which demonstrates that no intrinsic mechanism limits the association of up to four β-subunits within the $I_{Ks}$ complex.

*For correspondence: david.fedida@ubc.ca

**Competing interests:** The authors declare that no competing interests exist.

## Introduction

Kv7.1 (KCNQ1) is a voltage gated potassium channel expressed throughout the body. When paired with the accessory β-subunit, KCNE1, it has unique properties in several tissues, particularly the myocardium and inner ear. In the heart, KCNQ1 and KCNE1 together conduct the slow delayed rectifier current ($I_{Ks}$). This current is primarily responsible for stabilizing repolarization and shortening action potential duration at high heart rates (*Marx et al., 2002*; *Ackerman, 1998*). Several inherited mutations in KCNQ1 or KCNE1 result in the life-threating arrhythmogenic syndromes long QT (LQT) types 1 and 5, short QT type 2 or familial atrial fibrillation (*Splawski et al., 2000*; *Bellocq et al., 2004*; *Chen et al., 2003*). In addition to their role in the heart, KCNQ1 and KCNE1 are also involved in hearing, where mutations have been linked with Jervell and Lange-Nielsen syndrome, an inherited form of LQT syndrome accompanied by deafness (*Jervell and Lange-Nielsen, 1957*; *Neyroud et al., 1997*; *Schulze-Bahr et al., 1997*). KCNQ1 and KCNE1 complexes have also been found in the proximal tubule of the kidney, where they participate in secretory transduction (*Vallon et al., 2001*).

KCNQ1 has a classic six transmembrane domain structure consisting of a voltage sensor domain (VSD; S1-S4) and pore domain (S5-S6), which assemble into a tetrameric channel. Unlike other $K_v$ channels, KCNQ1 has flexible gating characteristics and the association with accessory β-subunits can drastically modulate its gating behavior (*Eldstrom and Fedida, 2011*; *Liin et al., 2015*). The best characterized of these accessory subunits is KCNE1, a single transmembrane domain protein proposed to reside in the exterior cleft formed between the voltage sensor domains of the α-

**eLife digest** The membrane that surrounds heart muscle cells contains specialized channels that can open and close to control the movements of charged ions into and out of the cell. This ion flow generates the electrical signals that stimulate the heart muscle to contract for each heart beat.

Different ion channels influence different steps in the initiation and termination of each electrical signal. For example, the $I_{Ks}$ ion channel complex helps to return the cell to a resting state so the heart muscle can relax. This allows chambers of the heart to fill with blood before the next beat pumps blood throughout the body. Mutations that affect $I_{Ks}$ cause serious heart conditions that affect heart rhythm, such as Long QT Syndrome.

The $I_{Ks}$ complex consists of channels that are each made of four copies of a protein called KCNQ1, through which potassium ions exit the cell. This channel opens in response to changes in the voltage across the cell membrane (known as the "membrane potential"). A small protein subunit called KCNE1 also makes up part of the complex, but it was not clear how many KCNE1 molecules combine with KCNQ1 to form a working channel complex. Several previous studies have reported two different results: that the KCNQ1 channel complex only exists with two KCNE1 molecules, or that the association is flexible, allowing the complex to contain up to four KCNE1 subunits.

Murray et al. have now constructed $I_{Ks}$ fusion channels out of different numbers of KCNQ1 and KCNE1 molecules to investigate how different KCNQ1:KCNE1 ratios affect how the channel works. Measuring the responses of these modified channels in mammalian cells revealed that channels with four KCNE1 subunits conducted ions better than channels with one or two KCNE1s. The channels containing fewer KCNE1s also opened at lower membrane potentials and after a shorter delay following a change in the membrane potential. Further experiments also supported the theory that up to four independent KCNE1 subunits may be easily added to the $I_{Ks}$ ion channel complex.

Murray et al. suggest that by being able to form channel complexes containing different numbers of KCNE1 subunits, cells can more flexibly control the rate at which ions flow out of the heart cells to tune the electrical signals that trigger each heart beat. The next challenges will be to determine the composition of the $I_{Ks}$ channel complex in adult heart cells and to investigate how the complex might change with disease.

subunits (*Sanguinetti et al., 1996*; *Barhanin et al., 1996*; *Osteen et al., 2010*; *Kang et al., 2008*; *Van Horn et al., 2011*). The association of KCNE1 has been found to increase channel conductance, remove inactivation, and depolarize the voltage dependence of activation. These actions have a profound effect on the rate of channel activation and its response to drugs and hormones (*Sanguinetti et al., 1996*; *Barhanin et al., 1996*; *Sesti and Goldstein, 1998*; *Yang and Sigworth, 1998*; *Pusch, 1998*; *Tristani-Firouzi and Sanguinetti, 1998*; *Nerbonne and Kass, 2005*; *Lerche et al., 2000*; *Yu et al., 2013*).

Despite the significant role of this ion channel complex in health and disease, the stoichiometry of the binding between KCNQ1 and KCNE1 remains controversial. Since the identification of the components that comprise this channel almost 20 years ago, several studies have reported a ratio of two β-subunits to every four α-subunits (*Kang et al., 2008*; *Wang and Goldstein, 1995*; *Chen et al., 2003*; *Morin and Kobertz, 2007*; *Morin and Kobertz, 2008*; *Plant et al., 2014*). However, other experiments have suggested a more flexible stoichiometry with anywhere from one to four β-subunits associating per KCNQ1 tetramer depending on their concentration (*Yu et al., 2013*; *Cui et al., 1994*; *Wang, 1998*; *Morokuma et al., 2008*; *Zheng et al., 2010*; *Wang et al., 2011*; *Strutz-Seebohm et al., 2011*; *Nakajo et al., 2010*). Most notably, two recent reports both utilizing total internal reflection fluorescence microscopy to count β-subunits by single molecule photobleaching reached opposing conclusions. Working in oocytes, Nakajo and colleagues characterized a flexible stoichiometry up to 4:4 that was dependent on the relative density of KCNE1 in the membrane (*Nakajo et al., 2010*). Using a similar approach in mammalian cells, *Plant et al. (2014)* reported that KCNQ1 tetramers were only present on the cell surface in a 2:4 or 0:4 stoichiometry with KCNE1. They also found that the 2:4 ratio was not exceeded even in the presence of increasing KCNE1 concentrations. Their observation of a maximum of 2 KCNE1 subunits per tetramer suggests

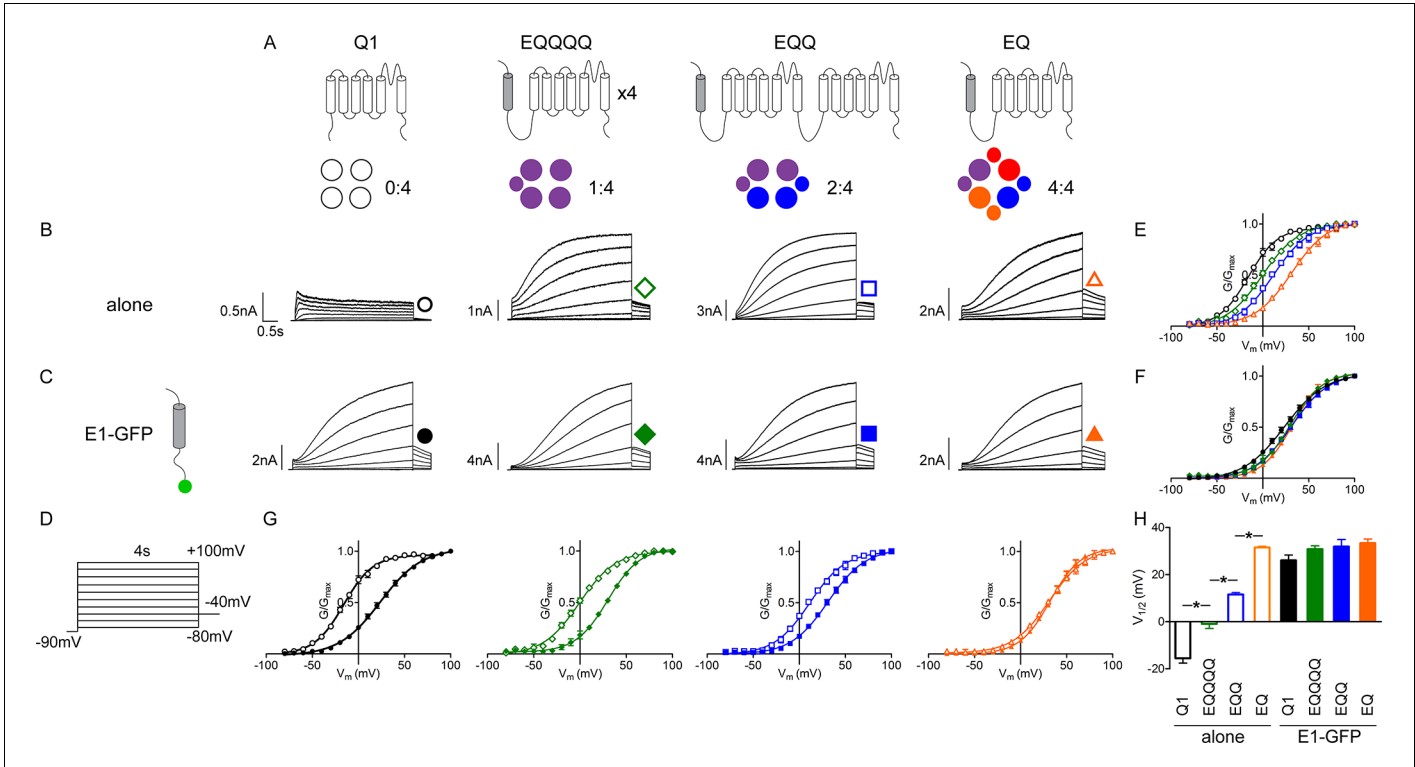

**Figure 1.** Additional β-subunits can alter $I_{Ks}$ channel complex gating properties. (A) Channel topology diagrams indicate the configuration and proposed stoichiometry of the channels. The KCNE1 sequences are shaded. Representative whole cell patch clamp recordings are shown for KCNQ1 (black circle), EQQQQ (green diamond), EQQ (blue square) and EQ (orange triangle), expressed alone (B; open symbols) or in combination with wild type KCNE1-GFP (C; filled symbols). (D) Currents were elicited using a 4 s isochronal activation protocol. Only odd numbered sweeps are shown for clarity. Tail current G-V plots are shown comparing the response to increasing number of β-subunits either by fusion (E) or by co-expression with KCNE1 (F). (G) G-V plots comparing each channel complex with and without co-expression of KCNE1. (H) Summary of each channel's $V_{1/2}$ of activation (n = 3–11; *p<0.05). See *Figure 1—source data 1*.

The following source data is available for figure 1:

**Source data 1.** $V_{1/2}$ of activation for $I_{Ks}$ channel complexes

an intrinsic cooperative mechanism preventing the association of a third or fourth β-subunit during channel assembly.

Here, we sought to clarify this issue using fusion channels where KCNE1 was linked to one (EQ - 4:4), two (EQQ – 2:4) or four (EQQQQ - 1:4) KCNQ1s. The channel complexes were evaluated in mammalian cells on their own or in combination with additional independent KCNE1 subunits using whole cell patch clamp, single-channel recordings and UV-crosslinking unnatural amino acid approaches. In each instance, we were able to demonstrate that all four of KCNQ1's exterior clefts are accessible to KCNE1, confirming the variable stoichiometry model for channel complex assembly. In addition, the single channel analysis suggests that a 4:4 stoichiometry of KCNQ1 to KCNE1 maximizes channel conductance.

## Results

### Characterization of $I_{Ks}$ channel constructs

To help differentiate possible channel complex stoichiometries, we created three $I_{Ks}$ fusion constructs (*Figure 1A*). The C-terminus of KCNE1 was connected with the N-terminus of KCNQ1 via a flexible linker (EQ), forcing a 4:4 stoichiometry of β and α subunits. An additional one or three KCNQ1 subunits were linked to the construct to make EQQ and EQQQQ, respectively (*Yu et al.,*

*2013*; *Wang, 1998*; *Nakajo et al., 2010*). A priori, EQQ, establishes a 2:4 KCNE1 to KCNQ1 subunit stoichiometry, and leaves two unoccupied clefts on each channel's exterior, while EQQQQ has a 1:4 ratio with three unoccupied clefts. KCNQ1, EQQQQ, EQQ and EQ were expressed independently in mammalian tsA201 cells and the resulting currents were characterized by whole cell patch clamp (*Figure 1B*). A 4 s isochronal activation protocol ranging from -80 to +100 mV was used to determine the $V_{1/2}$ of activation (*Figure 1D*). As is known, KCNQ1 expressed alone had a $V_{1/2}$ of -15.5 mV, while the presence of additional tethered β-subunits resulted in a progressive depolarization of the $V_{1/2}$ of activation to a maximum of ~+30 mV, (*Figure 1E and H*). This indicates that the linked subunits contribute to channel complex regulation.

Next, KCNQ1, EQQQQ, EQQ and EQ were co-expressed with a KCNE1-GFP construct. GFP was included in the construct to verify that KCNE1 was expressed and localized to the plasma membrane in each cell patched. The resulting currents were characterized using the same protocol (*Figure 1C*). In the presence of excess additional β-subunits, all the α-subunit complex configurations had $V_{1/2}$'s equivalent to that of EQ (*Figure 1F and H*). This convergence of the $V_{1/2}$'s of activation to ~+30 mV, particularly the depolarizing shift of EQQQQ and EQQ, suggests that the independent GFP-tagged β-subunits can participate in channel gating.

## Reduced single-channel conductance and latencies of EQQ and EQQQQ $I_{Ks}$ channel complexes

Current records in *Figure 2A* illustrate selected bursts of channel openings from the EQ, EQQ and EQQQQ (4:4, 2:4 and 1:4) channel complexes during 4 s depolarizations to +60 mV and repolarization to -40 mV to observe tail currents. The EQ 4:4 complexes open after a variable delay period and show fairly consistent bursts of opening that are close to 0.5 pA in amplitude. Similar single channel behavior is seen with KCNQ1+KCNE1, (*Werry et al., 2013*) as also shown in *Figure 3A*. In comparison, it can be seen that EQQ and EQQQQ channels open to lower levels than the EQ channels. In fact, EQQ channels show extremely long residence times in lower sublevels and rarely, or in the case of EQQQQ never, reach the fully open amplitudes seen in the EQ tracings. These observations are confirmed in the all-events amplitude histograms for the constructs, and the current-voltage relationships (*Figure 2B and C*). While for EQ it can be seen that the peak of the events amplitude histogram is at ~0.45 pA, it is only 0.18 pA for EQQ, and barely separable from the closed events distribution in the case of EQQQQ, despite very large numbers of events. The dotted line in the plotted current-voltage relationships (*Figure 2C*) refers to a slope conductance of 3.2 pS observed by *Werry et al. (2013)* from separately co-expressed KCNQ1 and KCNE1. Clearly, the data from the EQ construct overlay the prior published data, and the fit line gives a mean slope conductance of 3.0 pS. However, for EQQ, the mean conductance data gives a value of only 1.3 pS, well below that observed for KCNE1 and KCNQ1 co-expressed separately (*Figure 3C*) or as EQ together (p<0.01). Due to the small amplitudes and brevity of EQQQQ openings it was not possible to derive a conductance value.

In our prior analysis of wild type $I_{Ks}$, we described the presence of multiple sub-opening levels of the channel complex that were sometimes long-lived. Here we have analyzed single channel data from EQ and EQQ to compare their striking subconductance occupancy behavior (*Figure 2—figure supplement 2*). The all-points and idealized histograms are shown overlaying each other in *Figure 2—figure supplement 2*, and the tabulated data below. In EQ we observed the same five sublevels described before for E1 and Q1 expressed separately, plus an additional sixth smaller opening level at 0.08 pA that we did not see as clearly before. Visualizing this level was only possible due to improved signal to noise ratio in the present experiments. Again, EQ had its longest open residency in the second highest sublevel at 0.44 pA (36% of the time spent open), and split its remaining open time fairly evenly across the other sublevels. EQQ visited the same subconductance levels, but spent most of its time at the lower levels, only reaching the 0.44 pA level for 0.6% of the time it spent open. The most commonly occupied subconductance state for EQQ was the 0.29 pA sublevel, which it occupied 34% of the total open time. The data suggest that the open pore architecture of the $I_{Ks}$ channel complex is not altered by the presence of only two E1 subunits in EQQ, just that the stability of higher subconductance levels is reduced to such an extent that their occupancy becomes probabilistically unlikely. This is the explanation for the lower observed single channel conductance of EQQ (*Figure 2*), and likely EQQQQ as well.

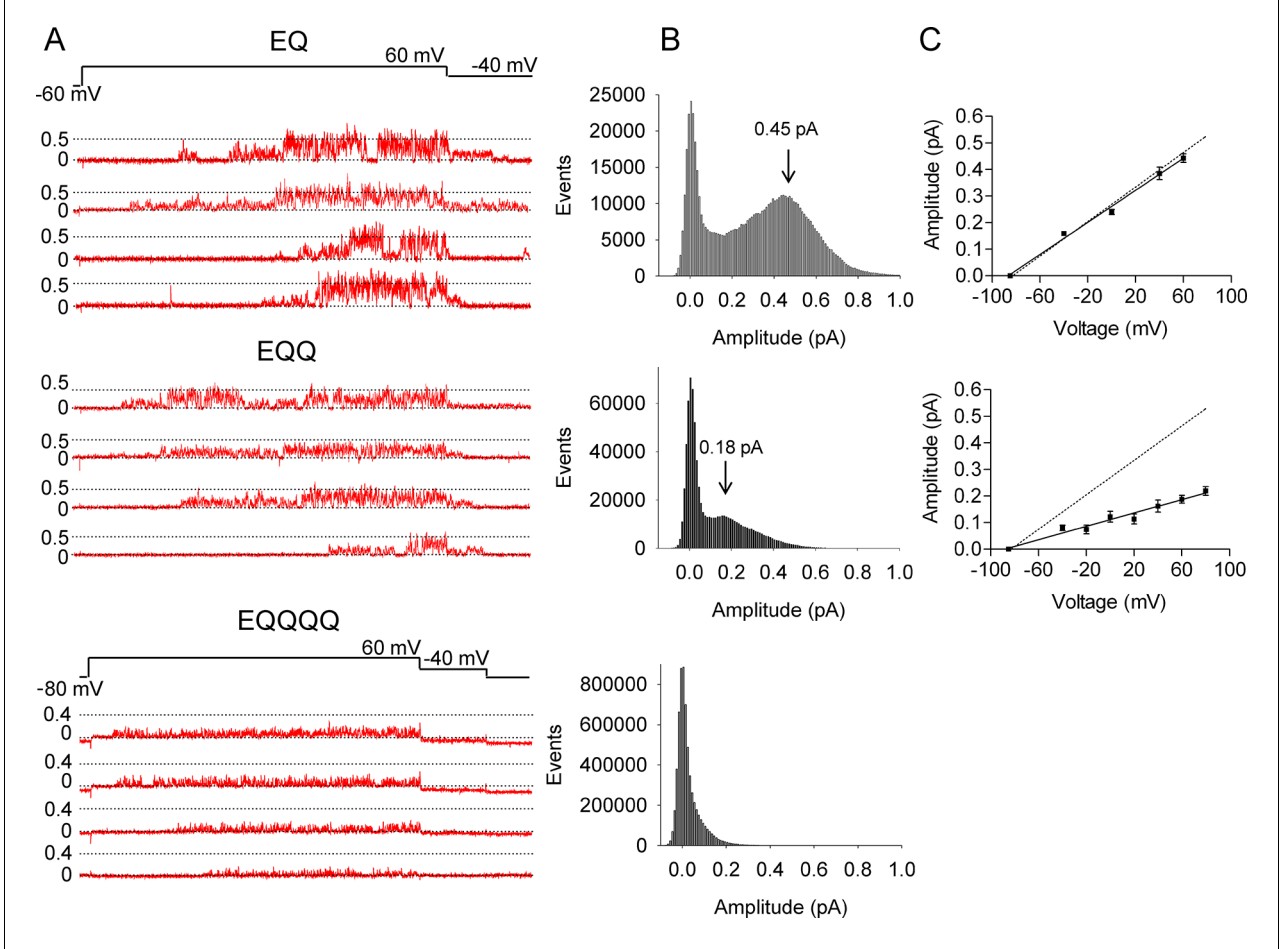

**Figure 2.** EQ, EQQ and EQQQQ show clear differences in single channel behavior. Membrane patches containing a single $I_{Ks}$ channel made up of EQ, EQQ, or EQQQQ were stepped from a holding potential of -60 mV (-80 mV for EQQQQ) to 60 mV for 4 s and then to -40 mV for 0.75 s as indicated in the protocol at top. (A) Representative traces of single channel recordings from cells expressing EQ (top), EQQ (middle) or EQQQQ (bottom). (B) All-points amplitude histograms of only the active single channel sweeps from a file of 100 sweeps, blank sweeps were removed to limit the 0 amplitude peak. All-points amplitude histograms containing all the blank sweeps are presented in *Figure 2—figure supplement 1*. For EQQQQ the histogram represents data from 3 cells. Arrows indicate the peak conductance as determined by Gaussian fits using Clampfit. (C) The peak open amplitudes for several voltages were plotted with an extrapolated K[+] channel reversal potential to derive a slope conductance for each construct. The slope conductance for $I_{Ks}$ made up of unlinked KCNE1 and KCNQ1, as previously published, (*Werry et al., 2013*) is indicated by a dashed line. Amplitudes for EQQQQ were too small to make analysis of conductance meaningful.

The following figure supplements are available for figure 2:

**Figure supplement 1.** All-points amplitude histograms.

**Figure supplement 2.** Subconductance analysis of EQQ and EQ demonstrates that as the number of KCNE1 subunits decreases so does higher conductance state occupancy.

We also found changes in the activation kinetics of EQQ and EQQQQ vs. EQ, as shown in *Table 1*. Consistent with more rapid activation of the EQQ and EQQQQ channel complexes, the first latency to opening of single channels was reduced compared with EQ, from 1.48 ± 0.18 s to 0.94 ± 0.07 s (p<0.05), and to 0.81 ± 0.07 s (p<0.01), respectively. These numbers may be compared with independently-expressed KCNE1 and KCNQ1, which shows a mean first latency of 1.50 ± 0.12 s (*Werry et al., 2013*) and (*Table 1*).

Interestingly, it was possible to increase the amplitude of EQQ and EQQQQ construct channel openings and delay their first latency to opening by co-expressing them with additional KCNE1

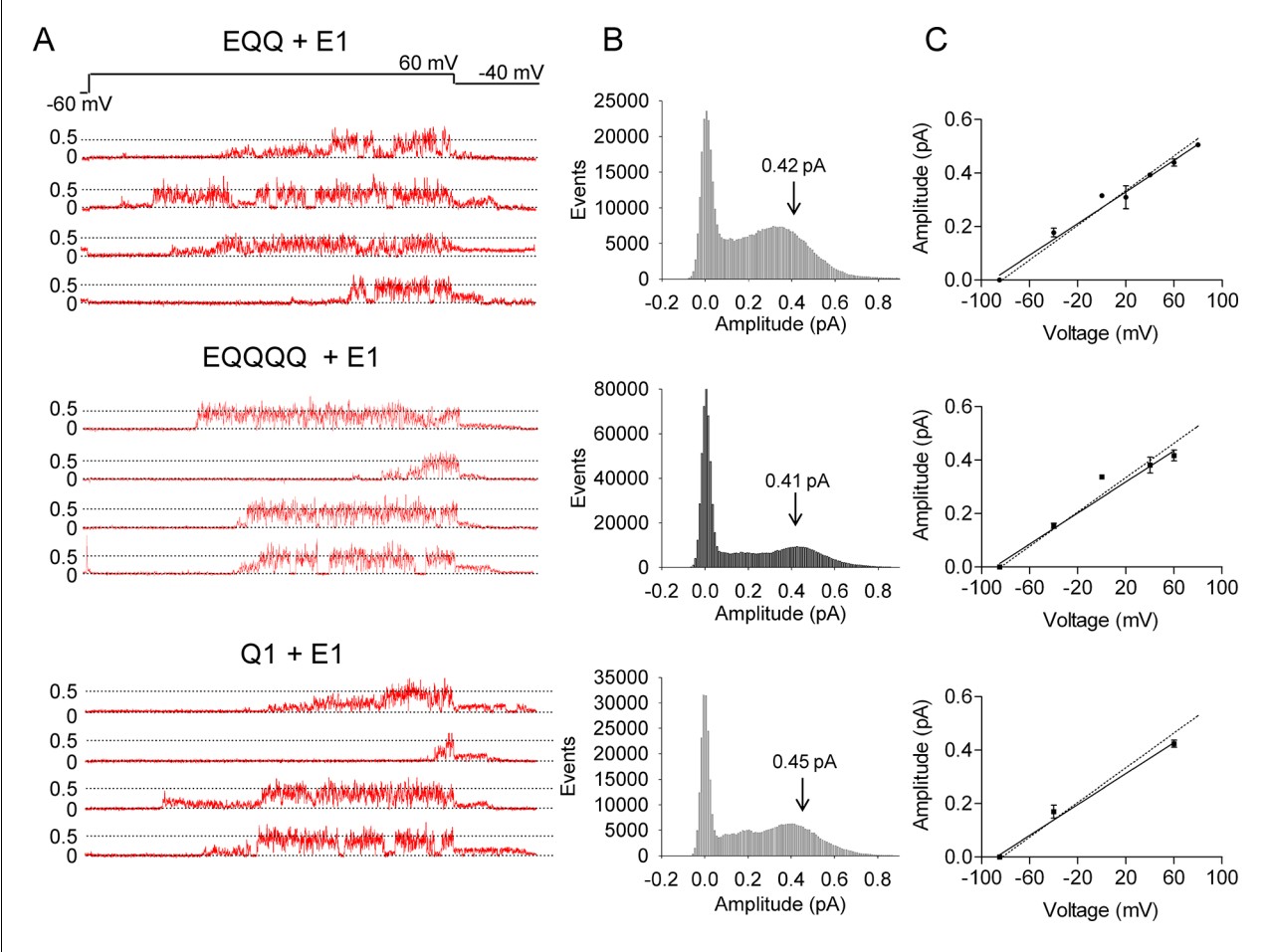

**Figure 3.** Co-expression of additional KCNE1 subunits restores wild type $I_{Ks}$ single channel behavior to EQQ and EQQQQ. (**A**) Representative traces of single channel recordings from cells expressing EQQ + KCNE1-GFP (top), EQQQQ + KCNE1-GFP (middle) or KCNQ1 + KCNE1-GFP (bottom). (**B**) All-points amplitude histograms of only the active single channel sweeps as described in *Figure 2*. Arrows indicate the peak conductance as determined by Gaussian fits using Clampfit. (**C**) The peak open amplitudes for several voltages were plotted as in *Figure 2*.

(*Figure 3*, and *Table 1*). These data are shown in a similar format and now it can be clearly seen that the current openings of the resulting EQQ+KCNE1 and EQQQQ+KCNE1 channel complexes routinely reach 0.4–0.5 pA, also confirmed by the open events histograms (*Figure 3B*) and the current-voltage relationships (*Figure 3C*). The slope of the current-voltage relationships for EQQ+KCNE1 and EQQQQ+KCNE1 were 3.0 ± 0.18 pS, and 2.9 ± 0.15 pS, respectively, which were not significantly different from KCNQ1 and KCNE1 expressed separately (2.9 ± 0.12 pS; P>0.05). The mean first latencies to opening of EQQ+KCNE1 and EQQQQ+KCNE1 were now 1.43 ± 0.08 s, and 1.44 ± 0.14 s, respectively, very similar to that reported before for KCNE1+KCNQ1 and measured again here, 1.50 ± 0.12 s.

## Expression and characterization of Bpa-incorporated $I_{Ks}$

To determine if the independently expressed β-subunits are directly interacting with the channel complex we employed a photo-crosslinking unnatural amino acid approach (*Farrell et al., 2005*; *Hino et al., 2005*; *Klippenstein et al., 2014*). *p*-Benzoylphenylalanine (Bpa) can form a covalent crosslink following excitation with UV light (350–380 nm) (*Figure 4A*). The UV light excites the side chain's central ketone, generating a triplet oxygen radical. This short-lived radical can abstract a hydrogen atom from a carbon atom within a 3.1-angstrom radius. The resulting alkyl radical then rapidly reacts with the remaining ketyl radical in Bpa to form a covalent bond (*Dorman and*

*Prestwich, 1994*). Bpa was incorporated into the KCNE1 sequence using the amber stop codon (TAG) suppression system (*Figure 4A*) (*Ye et al., 2008*; *Xie and Schultz, 2005*; *Young and Schultz, 2010*; *Chin and Schultz, 2002*). F57 was selected for mutation as it is located within the KCNE1 transmembrane domain and has been previously suggested to interact with KCNQ1 in the channel's closed state although no physical association has been established (*Melman et al., 2001*; *Chen and Goldstein, 2007*). Methionine 62 was also mutated to a tryptophan as a precaution against an inappropriate start site downstream of the TAG site. The M62W KCNE1 mutation has been previously shown not to impact channel gating, (*Chen and Goldstein, 2007*) and the same lack of effect was observed here (*Figure 4—figure supplement 1*).

To evaluate the effect of Bpa incorporation on $I_{Ks}$ channel gating, the F57TAG/M62W KCNE1-Y40TAG GFP (F57Bpa KCNE1) construct was co-expressed with KCNQ1 and the tRNA and amino-acyl-tRNA-synthetase (RS) pair in media supplemented with 1 mM Bpa. This resulted in characteristic $I_{Ks}$ currents indicating the successful rescue expression of full length KCNE1 (*Figure 4B*, upper). Plots of tail current G-V relations demonstrated that F57Bpa KCNE1 showed only a 2 mV depolarizing shift in the $V_{1/2}$ of activation compared to wild type $I_{Ks}$ (wild type, 26.1 ± 2.2 mV vs. F57Bpa, 28.2 ± 6.8 mV; n = 3–11) (*Figure 4C*). The same was found for F57Bpa KCNE1 co-expressed with EQQQQ, EQQ and EQ channels compared to wild type KCNE1 (*Figure 4—figure supplement 2*). This demonstrates that incorporation of Bpa at position 57 does not significantly alter expression or channel gating properties.

In the absence of Bpa, translation is terminated at the TAG codon, resulting in the expression of a truncated 1–56 β-subunit (*Figure 4B*, lower). Currents in the absence of Bpa were similar to KCNQ1 alone. This indicated that there was good fidelity at the TAG site and that the truncated KCNE1 had no significant impact on the regulation of the channel complex.

To further validate Bpa incorporation, Western blot analysis was also performed (*Figure 4D*). In the absence of Bpa, no full-length KCNE1 was observed (lanes 2 and 4). Full-length expression of KCNE1-GFP was observed in the presence of Bpa for both the F57TAG KCNE1-Y40TAG GFP and wild type KCNE1-Y40TAG GFP constructs indicating good suppression of the amber stop codons by the tRNA and RS. The lower bands present in lanes 2 and 3 represent the truncated expression of KCNE1-Y40TAG GFP. In the absence of Bpa (lane 2), all translation ends at the Y40 stop codon resulting in a 20 kDa product that does not fluoresce. In the presence of Bpa, individual cells transfected with all 3 plasmids (KCNE1, tRNA, RS) have the complement to suppress the amber stop codon and incorporate Bpa, resulting in full-length expression (lane 3, upper band). Some cells will be transfected with the KCNE1-Y40TAG GFP plasmid but fail to receive the tRNA and/or RS. These cells cannot suppress the stop codon and express the truncated form (lane 3, lower band).

## UV-dependent channel crosslinking

To determine if F57Bpa directly interacts with KCNQ1, a 300 ms flash of UV light (365 nm) was applied while holding at -90 mV followed by a 4 s activation step (+60 mV) and repeated with each successive sweep (*Figure 5A*). The application of UV light to F57Bpa channels in the closed state resulted in a rapid and complete loss of peak current (*Figure 5B*-left). A diary plot of the normalized

**Table 1.** Summary of single channel parameters.

| Construct | First latency (s) | n (cells) | # Active sweeps | Conductance (pS) | n (cells) |
|---|---|---|---|---|---|
| KCNQ1 + KCNE1 | 1.50 ± 0.12 | 3 | 71 | 2.9 ± 0.12 | 2 |
| EQQQQ | 0.81 ± 0.07 | 3 | 124 | n.d. | |
| EQQ | 0.94 ± 0.07 | 3 | 128 | 1.3 ± 0.01 | 3-6 |
| EQ | 1.48 ± 0.18 | 3 | 36 | 3.0 ± 0.11 | 2-4 |
| EQQ + KCNE1 | 1.43 ± 0.08 | 4 | 178 | 3.0 ± 0.18 | 2-5 |
| EQQQQ + KCNE1 | 1.44 ± 0.14 | 3 | 52 | 2.9 ± 0.15 | 1-4 |

P values for first latency: EQQ vs. E1+Q1, EQ, EQQQQ+E1 and EQQ+E1 p<0.05; EQQQQ vs. E1+Q1, EQ, EQQQQ+E1 and EQQ+E1 p<0.05. P values for conductance: EQQ vs. Q1+E1, EQQQQ+E1, EQ and EQQ+E1 p<0.001. Not determined (n.d.).

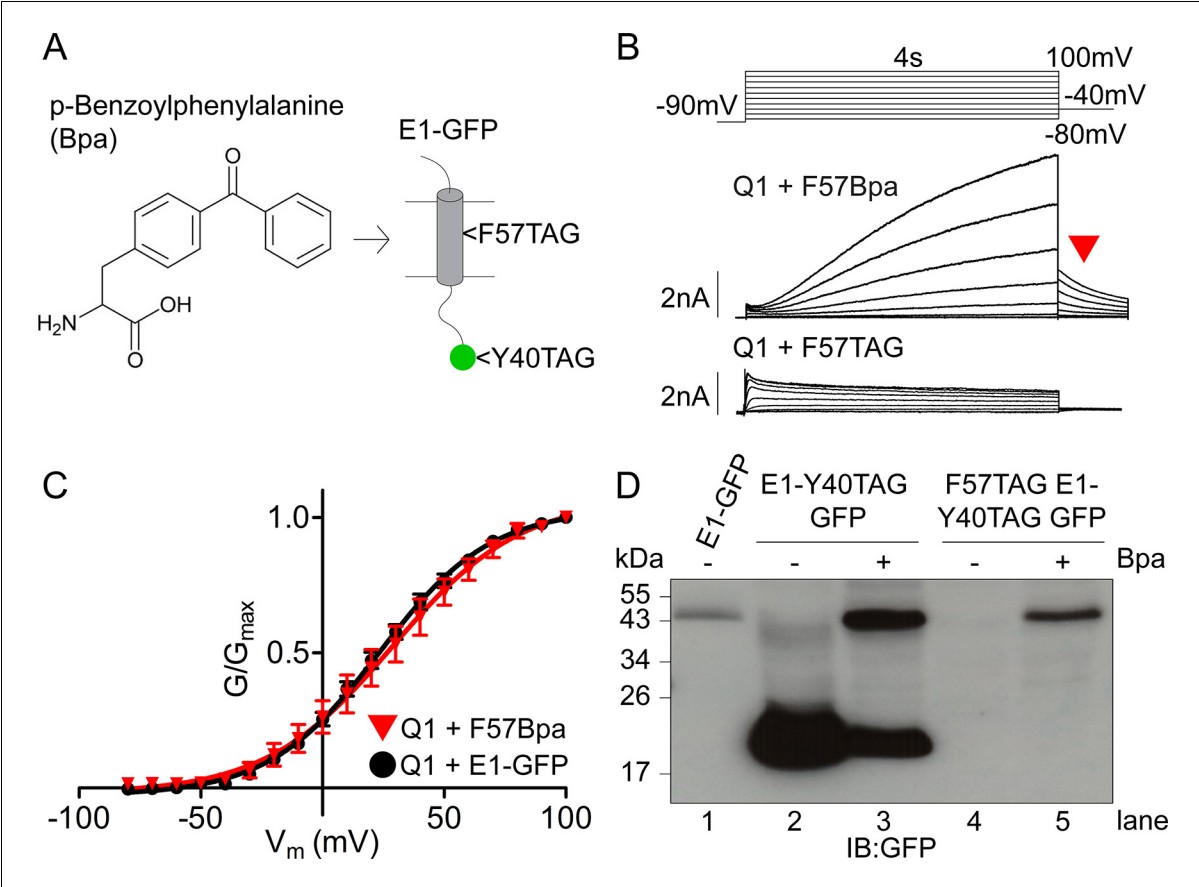

**Figure 4.** Expression and characterization of F57Bpa KCNE1 in $I_{Ks}$. (**A**) Schematic showing the structure of the UV- activated crosslinking amino acid, Bpa, and the position of F57 within the transmembrane domain of KCNE1. Bpa was also incorporated at position 40 in the GFP sequence. (**B**) Currents were obtained using the isochronal activation protocol from F57TAG/M62W KCNE1-Y40TAG GFP constructs cultured in the presence (upper) or absence (lower) of 1 mM Bpa. Only odd numbered sweeps are presented for clarity. (**C**) G-V relations are shown from wild type (black circles) and F57Bpa KCNE1 $I_{Ks}$ (red triangles) (n = 3–11). (**D**) Western blot depicting the expression of wild type KCNE1-GFP, KCNE1-Y40TAG GFP and F57TAG/M62W KCNE1- Y40TAG GFP constructs in the presence and absence of 1 mM Bpa.

The following source data and figure supplements are available for figure 4:

**Source data 1.** $V_{1/2}$ of activation for $I_{Ks}$ channel complexes using 31 aa (sub S) or 52 aa (sub L) linker
**Figure supplement 1.** M62W KCNE1 does not alter $I_{Ks}$ channel activation compared to wild type.
**Figure supplement 2.** F57Bpa KCNE1 does not alter channel gating of KCNQ1, EQQ and EQ compared to wild type KCNE1.

peak currents shows the establishment of a stable baseline (five sweeps) followed by the UV-dependent inhibition of current (*Figure 5B*-right). Co-expression of KCNQ1 with wild type KCNE1 also exhibited a UV-dependent progressive decline, or rundown, in peak current (*Figure 5B*-right, open symbols). This relatively slow loss of current was observed equally across all $I_{Ks}$ channel configurations (*Figure 5—figure supplement 1*); The Bpa-specific UV-crosslinking effect was found to be much faster and more complete than the wild type rundown (*Figure 5B*).

To further confirm covalent crosslinking between F57Bpa KCNE1 and KCNQ1, co-immunoprecipitation was performed (*Figure 5—figure supplement 2*). F57Bpa or wild type KCNE1-GFP was co-expressed with KCNQ1 and exposed to UV light. Western blot analysis indicated a clear increase in the co-association of F57Bpa KCNE1-GFP with KCNQ1 in the presence of longer UV irradiation compared to wild type KCNE1-GFP. These findings, in combination with the UV-current recordings,

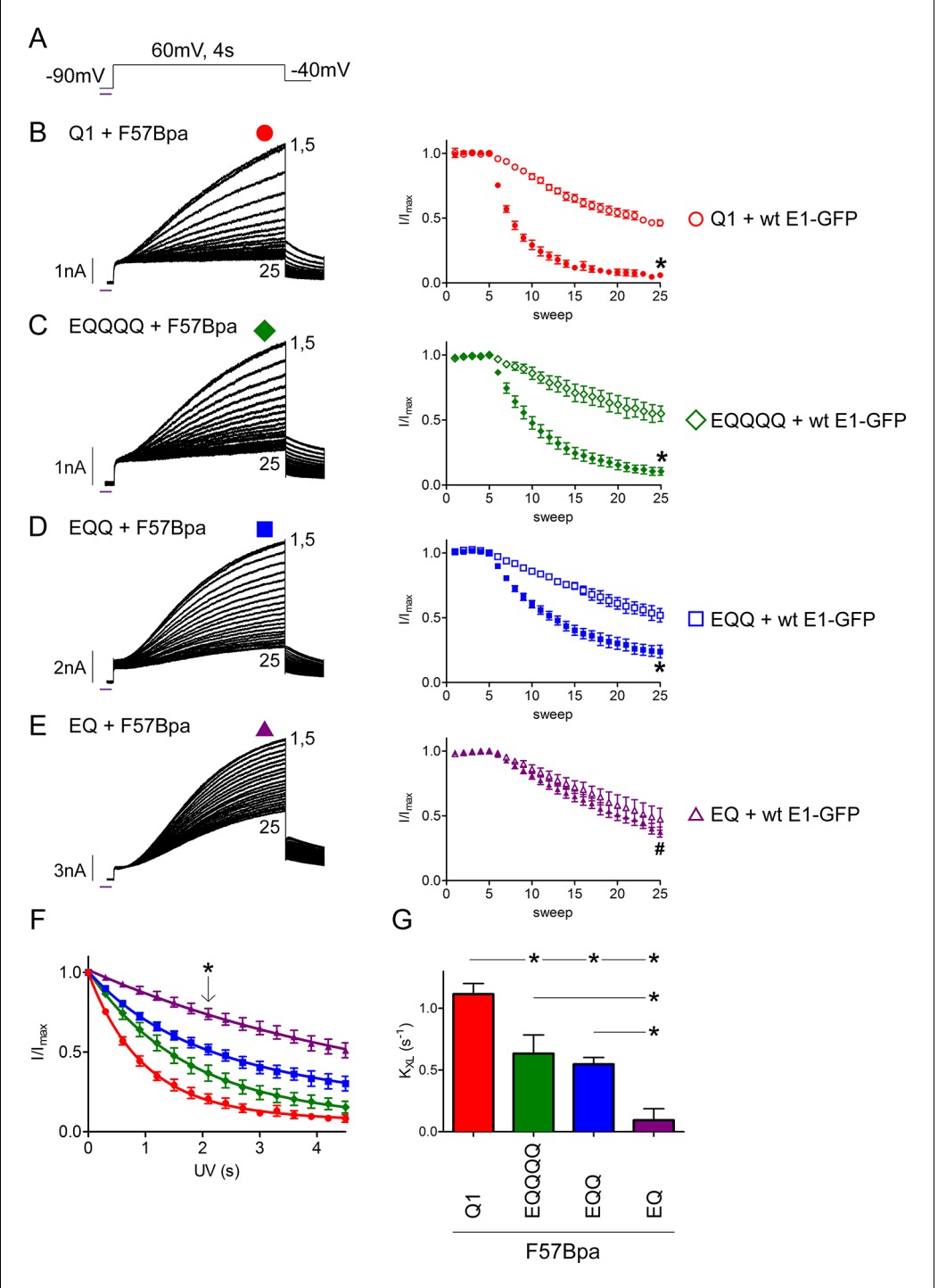

**Figure 5.** The $I_{Ks}$ channel complex does not have a restricted β-subunit stoichiometry. (**A**) Schematic of the UV-voltage protocol. A flash of UV light (purple line) is applied once per sweep for 300 ms at -90 mV before a 4s activation step to +60 mV. Representative currents are shown for KCNQ1 (**B**), EQQQQ (**C**), EQQ (**D**) and EQ (**E**) co-expressed with F57Bpa KCNE1 (left). For all recordings, UV was applied at sweep 6 after a stable baseline had been established. Sweeps 1 and 5–25 are presented. Diary plots (right) of the UV treated normalized peak currents for each channel construct co-expressed with F57Bpa KCNE1 (filled symbols) or wild type KCNE1 (open symbols) (n = 3–7; *p<0.01; #p>0.05). (**F**) Plots of the normalized peak current vs. cumulative UV exposure for KCNQ1 + F57Bpa KCNE1 (red circle), EQQQQ + F57Bpa KCNE1 (green diamond), EQQ + F57Bpa KCNE1 (blue square) and EQ + F57Bpa KCNE1 (purple triangle). *p<0.05 comparing normalized peak currents after 2.1 s of UV exposure

*Figure 5 continued on next page*

*Figure 5 continued*

(n = 5–7). (G) Summary of the crosslinking rates obtained from each cell (See *Figure 5—source data 1*) (n = 5–7; *p<0.05).

The following source data and figure supplements are available for figure 5:

**Source data 1.** Crosslinking rate constants for $I_{Ks}$ channel complexes.

**Source data 2.** Rate constants for $I_{Ks}$ channel complex rundown.

**Figure supplement 1.** UV-rundown is consistent across all the $I_{Ks}$ channel configurations.

**Figure supplement 2.** F57Bpa KCNE1 covalently crosslinks with KCNQ1.

indicate that F57Bpa forms a UV-dependent covalent crosslink with KCNQ1 that traps the channels in the closed state.

The EQQ and EQQQQ channel complexes have two and three voltage sensor clefts that are not occupied by β-subunits, respectively. The suggestion that $I_{Ks}$ can only adopt a 2:4 stoichiometry implies that access to the unoccupied clefts is restricted, while in a variable stoichiometry model they would be accessible (*Plant et al., 2014*; *Nakajo et al., 2010*). Since F57Bpa interacts with KCNQ1 in the channel's closed state, we used this observation to test whether an independently expressed F57Bpa KCNE1 β-subunit could enter an open cleft in EQQQQ or EQQ and crosslink with the channel. UV irradiation of EQQQQ or EQQ co-expressed with F57Bpa KCNE1 resulted in an accelerated rate of peak current decay compared to expression of these constructs with wild type KCNE1-GFP (*Figure 5C and D*). The results clearly suggest that F57Bpa can access unoccupied clefts in the EQQQQ and EQ constructs to allow UV-crosslinking to occur.

We also considered that an independently expressed β-subunit could displace one of the tethered β-subunits in the channel complex and crosslink with KCNQ1. To test this possibility, we co-expressed F57Bpa KCNE1 with the EQ fusion construct. The EQ channel complex is proposed to have all the voltage sensor clefts occupied. UV-treatment did not cause any crosslinking-dependent current decrease compared to EQ co-expressed with wild type KCNE1 (*Figure 5E*). This finding indicates that there is minimal displacement of the tethered β-subunits in a voltage sensor cleft by an independent one.

To evaluate the differences in crosslinking, the mean rate constant ($K_{XL}$) was determined by double exponential decay fit of the normalized peak current vs. cumulative UV exposure for each of the replicates, where the slow rate constant was set to the value obtained from the wild type rundown ($K_{RD}$) (*Figure 5F and G* and Materials and methods). Analysis of the fast rate constants revealed more rapid F57Bpa KCNE1 crosslinking as the number of potentially available clefts increased (EQQ – 2 clefts, $0.55 \pm 0.06$ s$^{-1}$; EQQQQ – 3 clefts, $0.63 \pm 0.06$ s$^{-1}$; KCNQ1 – 4 clefts, $1.12 \pm 0.09$ s$^{-1}$ (n = 5–7)) (*Figure 5G*). These findings demonstrate that more F57Bpa KCNE1-GFP subunits are interacting with the KCNQ1 channel complexes, as more clefts are available. EQ, which has no available clefts, had a UV decay rate no different than the rundown observed for all wild type channels, and significantly slower than other channel configurations ($K_{XL}$: EQ + F57Bpa, $0.09 \pm 0.09$ s$^{-1}$: p<0.01 (n = 5), *Figure 5G*). The intermediate rates of crosslinking for EQQQQ and EQQ between KCNQ1 and EQ indicate that F57Bpa KCNE1 can access the unoccupied clefts in the channel complexes. These findings demonstrate that there is no intrinsic restriction of β-subunit association and confirms a variable stoichiometry model for $I_{Ks}$ channel complex composition.

## Discussion

Our study is the first to apply single-channel analysis and unnatural amino acid substitution to address the question of β-subunit stoichiometry within the $I_{Ks}$ channel complex. Using these approaches we were able to determine that up to four KCNE1's can interact per KCNQ1 tetramer. Whole cell patch clamp studies indicated that EQQQQ and EQQ have a hyperpolarized voltage dependence of activation compared to EQ or wild type $I_{Ks}$, which could be normalized by co-expression of additional KCNE1. We also observed the same phenomenon in single channels, where

EQQQQ and EQQ had a shorter first latency to open and reduced conductance that equaled EQ and wild type $I_{Ks}$ channels when KCNE1 was co-transfected. Finally, our key experiment using a photo-crosslinking unnatural amino acid, unequivocally demonstrated that KCNE1 can incorporate into the EQQQQ and EQQ channel complexes. Taken together, our findings verify that up to four KCNE1 subunits can regulate KCNQ1.

As summarized in the Introduction, several previous studies support these findings, in suggesting that the ratio is variable ranging from 1–4 KCNE1's depending on their availability. However, this debate has recently coalesced around an opposing idea that $I_{Ks}$ has a strict 2:4 ratio of KCNE1's: KCNQ1's (*Kang et al., 2008*; *Morin and Kobertz, 2008*; *Plant et al., 2014*). This interaction has been investigated using a variety of approaches, including macroscopic currents, pharmacology and most recently by total internal reflection fluorescence microscopy. The report by *Plant et al. (2014)* utilizing single molecule photo-bleaching counted a maximum ratio of 2 β-subunits per KCNQ1 tetramer in opposition to a previous study using a similar approach (*Nakajo et al., 2010*). They suggest a possible explanation for the different conclusions was their use of mammalian cells, representing a better model system than *Xenopus* oocytes for $I_{Ks}$ assembly. Our findings using three different techniques in mammalian cells do not support this difference. We clearly show that an independently expressed KCNE1 can associate with EQQ, so that a stoichiometry greater than 2:4 is possible, and indeed that there is no intrinsic mechanism limiting the association of KCNE1 with the channel complex.

In the current study we utilized fusion proteins, different versions of which have been used previously. It is notable that the use of these proteins has not always yielded the same result. For example, in the first report of $I_{Ks}$ fusion channels, no significant difference between EQQ and EQ channels was found, although a slowing of activation kinetics was observed when EQ and KCNE1 were co-expressed (*Wang et al., 1998*). A similar result was also found using EQ and EQQ variants that had 30 mV hyperpolarizing shifts in their $V_{1/2}$ (*Chen et al., 2003*). More recently, two other studies have reported differences in the current-voltage relationship between EQ and EQQ that are consistent with our findings (*Yu et al., 2013*; *Nakajo et al., 2010*). *Nakajo et al. (2010)* also reported a small depolarizing shift in the $V_{1/2}$ when EQQ was co-expressed with KCNE1 in oocytes. One possible explanation for the differences in behavior could be the composition of the KCNE1-KCNQ1 linker in the fusion proteins. The earlier reports appear to use constructs where the C- and N-termini of KCNE1 and KCNQ1 are directly fused while later studies have inserted flexible linkers. Here, we constructed channels containing 52 and 31 amino acid linkers both of which exhibited a difference in $V_{1/2}$ between EQQ and EQ, and when EQQ was co-expressed with KCNE1 (*Figure 1* and *Figure 4—figure supplement 2*). This is consistent with the other longer E1-Q1 linker studies and suggests that restricting the C- (KCNE1) or N-termini (KCNQ1) may have subtle effects on channel gating that were not observed in basic macroscopic characterizations.

## KCNQ1 regulation by KCNE1

While determining the subunit stoichiometry was our primary goal, there were some additional observations of interest for KCNE1's role in modulating channel gating. The $V_{1/2}$'s of the current-voltage relationships for KCNQ1 alone, EQQQQ, EQQ, and EQ, increased linearly with increasing KCNE1 presence (*Figure 6*), as can also be observed using the data presented by *Yu et al. (2013)*. Each β-subunit that associates appears to have a similar contribution in regulating the midpoint of the voltage dependence of activation, and the linear change in the $V_{1/2}$ does not support cooperative gating effects of additional KCNE1 association.

The results of our crosslinking study also implicate KCNE1 in regulating the pore domain by showing that residue F57 in KCNE1 interacts with KCNQ1 in the resting state in a conformation that prevents channel opening (*Figure 5*). F57 has been identified as part of the 'activation triplet' consisting of F57, T58 and L59, which has been found to interact with the pore region of the channel (*Wang et al., 1996*; *Tapper and George, 2001*; *Melman et al., 2004*). A crosslinking event between KCNE1 and the pore region could restrict its movement and prevent channel opening. There are several theories for how KCNE1 delays channel opening, including that KCNE1 inhibits the movement of the S4 domain in the voltage sensor, (*Nakajo and Kubo, 2007*; *Ruscic et al., 2013*) or that it slows the opening of the pore domain (*Rocheleau and Kobertz, 2008*; *Osteen et al., 2010*; *Barro-Soria et al., 2014*). Our findings suggesting that KCNE1 potentially impacts both are supported by a recent paper by *Zaydman et al. (2014)*. They propose that KCNE1 does not directly

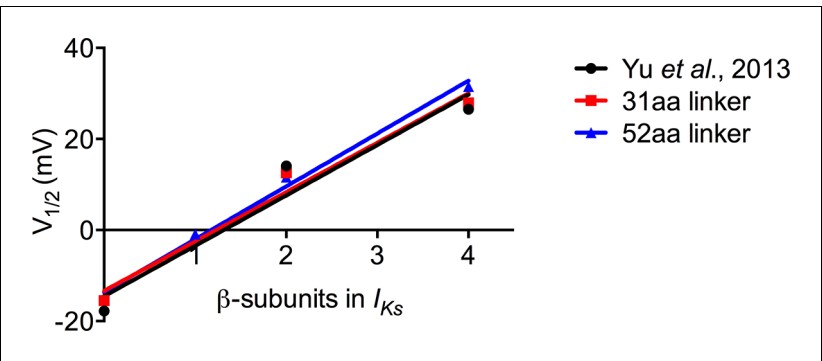

**Figure 6.** Increasing KCNE1 has linear effect on depolarizing shift in the $V_{1/2}$ of activation. Plot of the mean tail current G-V relations comparing KCNQ1 alone (0:4), EQQQQ (1:4), EQQ (2:4) and EQ (4:4) for fusion proteins with 31 amino acid and 52 amino acid linker as well as mean data obtained from *Yu et al. (2013)*. $R^2$ values: Yu et al, 0.94; 31aa, 0.97; 52aa, 0.99.

affect VSD activation or pore opening, and instead suggest that KCNE1 alters the state-dependent interactions linking the VSD and pore in KCNQ1 channels. If KCNE1 affects the interactions between the two domains, it is plausible that KCNE1 not only independently modulates the voltage dependence of its own α-subunit, but also participates in interactions that, when covalently crosslinked, can prevent channel opening. Clearly additional studies are required to better understand the detailed nature of these interactions.

## Implications of the single channel studies

One consequence of the prolonged debate over $I_{Ks}$ stoichiometry has been to suppress investigation of sub-saturating KCNE1 on the regulation of KCNQ1. Since the nature of the channel complex has been in doubt, it has been difficult to consider the effects of a partially chaperoned channel. By confirming that $I_{Ks}$ has a variable stoichiometry up to 4:4, our analysis of EQQ provides some insights into the channel's regulation. In addition to a hyperpolarizing shift in $V_{1/2}$ (*Figure 1*), our single channel analysis indicated that EQQ and EQQQQ channels had a reduced first latency to open at +60 mV compared with wild type (*Table 1*). Also, unsaturated channels rarely reach a fully conducting state, and instead they primarily reside in subconducting conformations (*Figures 2*, and *Figure 2— figure supplement 2*). The modulation of both these parameters is consistent with the kind of regulatory model proposed by *Zaydman et al. (2014)*, in which KCNQ1 has intermediate and activated voltage sensor states, both of which lead to channel opening, but where KCNE1 suppresses opening in the intermediate state. In this scenario, EQQ and EQQQQ would represent hybrid channels; QQ subunits lacking E1 could open from an earlier gating state while those associated with E1 open from a more fully activated state. This would change the interacting forces on the subunits, move channel opening to an earlier conformation in the activation pathway, and likely left-shift the $V_{1/2}$ of activation and reduce the first latency for opening as we observed.

The same hybrid characteristics can be applied to the behavior of the pore. We have previously observed that wild type channels show a mean conductance of 3.2 pS and also exhibit several subconducting levels (*Werry et al., 2013*). On its own, KCNQ1 is thought to have a reduced channel conductance, although this has not been confirmed by single channel analysis (*Yang and Sigworth, 1998*; *Pusch, 1998*). Our findings indicate that in the presence of only two KCNE1 subunits, the pore is still capable of achieving a wild type conducting state, but much less frequently such that a mean conductance of ~1.3 pS was observed. In the presence of only one E1 subunit (EQQQQ) the effect was even greater and points to a destabilized open pore configuration in the absence of KCNE1. Others have reported that KCNE1 can interact and modulate the characteristics of the pore domain but this is the first direct demonstration that sub-saturating levels of KCNE1 lowers mean single channel conductance, and it indicates that 4 KCNE1's preferentially stabilize the ion throughput rate of the open pore (*Wang et al., 1996*; *Tapper and George, 2001*; *Melman et al., 2004*). These findings indicate that a full complement of KCNE1 enhances KCNQ1 channel activity.

## Stoichiometry of the $I_{Ks}$ channel complex in cardiomyocytes

While the majority of studies have focused on the composition of $I_{Ks}$ in model systems, the stoichiometry on the surface of native cardiomyocytes has not yet been specifically addressed. The implication of variable stoichiometry suggests a distribution of stoichiometries depending on the expression level (*Nakajo et al., 2010*). A characterization of $I_{Ks}$ currents in dog cardiac myocytes determined the $V_{1/2}$ at 37°C for $I_{Ks}$ across different regions of the ventricular wall to be ~25 mV (*Liu and Antzelevitch, 1995*). Additionally, co-localization studies of KCNE1 and KCNQ1 in rat cardiomyocytes indicated that KCNE1 is present in excess (*Wang et al., 2013*). Our results and these previous findings suggest that under these conditions, it is likely that many of the $I_{Ks}$ channel complexes have a β-subunits configuration exceeding 2:4.

A variable stoichiometry model for $I_{Ks}$ has some interesting implications. A fully saturated channel has a depolarizing shift in the current-voltage relationship, extremely slowed activation and increased conductance (*Figure 1* and *2*). However, during a normal cardiac action potential this would result in limited activation. If the stoichiometry can be regulated based on availability, (*Nakajo et al., 2010*) then altering the expression could change the distribution of channel composition, shifting the overall activation kinetics of the current. Additionally, there is evidence for the association of other β-subunits. All five of the KCNE genes (KCNE1-5) can be expressed in cardiomyocytes (*Lundquist et al., 2006*; *Bendahhou et al., 2005*; *Radicke et al., 2006*). Each of these accessory proteins has been found to alter KCNQ1 channel gating in distinct ways (*Eldstrom and Fedida, 2011*; *Liin et al., 2015*). There is also potential for different KCNE's associating with the same channel to modulate its function (*Manderfield and George, 2008*; *Jiang et al., 2009*). Flexibility in KCNQ1's β-subunits association, both in number and in type, allows for a powerful mechanism to modulate repolarization in the heart through changes in expression.

## Conclusion

The number of β-subunits present in the $I_{Ks}$ channel complex is critical for its regulation. Here, we have shown that up to four KCNE1 subunits can associate with the KCNQ1 tetramer. Variable association allows for greater flexibility in the modulation of $I_{Ks}$ current but also potentially increases the heterogeneity of channel species in the membrane. The development of models of cardiac function, $I_{Ks}$ channel structure and activity, pharmaceutical screens and transgenic animals should include consideration of this potential diversity in $I_{Ks}$ channel configuration to best reflect the underlying physiology.

## Materials and methods

### Reagents

Bpa was obtained from Bachem. All other reagents and chemicals were from Sigma-Aldrich (St. Louis, MO, USA).

### Molecular biology

A human KCNE1-Flag-GFP pcDNA3 construct was mutated using the QuikChange II Site-Directed Mutagenesis Kit (Agilent Technologies, Santa Clara, CA, USA) using the following primers F57TAG 5'-ggtactgggattcttcggcttc<u>tag</u>accctgggcatc-3', M62W 5'-cttcaccttgggcatc<u>tgg</u>ctgagctac-3' and Y40TAG-GFP 5'-agggcgatgccacc<u>tag</u>ggcaagctg-3' according to the manufacturer's protocol. The EQ, EQQ and EQQQQ constructs were generated by linking the C-terminus of KCNE1 with the N-terminus of one, two or four human KCNQ1 sequences. The linker between KCNE1 and KCNQ1 consisted of 31 amino acids including a flag and V5 epitopes (UV experiments) or 52 amino acids, which included an additional 21 aa (SRGGSGGSGGSGGSGGSGGRS) inserted after the flag sequence (whole cell and single channel experiments). For EQQQQ and EQQ, the linker between KCNQ1's was 22 amino acids including a V5 tag. All mutations were confirmed by sequencing.

### Cell culture and transfections

tsA201 transformed human embryonic kidney (HEK) 293 or *ltk-* mouse fibroblast (LM) cells were grown in Minimum Eagle Medium (Thermo Fisher Scientific, Waltham, MA, USA) supplemented with

10% fetal bovine serum (Thermo Fisher Scientific), 100 U/ml penicillin and 100 ug/ml streptomycin (Thermo Fisher Scientific). Cells were maintained at 37°C in a humidified atmosphere containing 5% $CO_2$. One day prior to transfection, cells were lifted with 1 min exposure to trypsin/EDTA and re-plated on 25 $mm^2$ glass coverslips. $I_{Ks}$, wild type or Bpa-incorporated, channels were over-expressed by transient transfection using Turbofect (Thermo Fisher Scientific) in tsA201 cells according to the manufacturer's protocol in the presence of media supplemented with 1 mM Bpa. In some cases, a Y40TAG-GFP construct was used to indicate a successful transfection. The F57TAG KCNE1-Y40TAG GFP, KCNQ1 (or EQQQQ/EQQ/EQ), were co-transfected with a mutated tRNA, and RS pair; (*Ye et al., 2008*) in a 5:1:1:1 ratio, while wild type KCNE1-GFP and KCNQ1 (or EQQ/EQ) plasmids were transfected in a 3:1 ratio. For transfections lacking Bpa, wild type GFP was used at a ratio of 0.05 to indicate transfected cells. For single channel recordings constructs were transfected in LM cells using Lipofectamine2000 (Thermo Fisher Scientific). All recordings were performed 24–48 hr post-transfection. Adherent cells on coverslips were removed from their media, washed with external solution and transferred to the recording chamber containing external solution. All recordings were performed at room temperature.

In preliminary rescue experiments, there were occasional instances of GFP positive cells in the absence of Bpa. To ensure accurate reporting of full length KCNE1 expression, a TAG site was also introduced at position 40 in the GFP sequence. The F57TAG/M62W KCNE1 – Y40TAG GFP construct was used in all experiments (abbreviated to F57Bpa KCNE1) unless otherwise indicated.

## Patch-clamp electrophysiology

Whole-cell current recordings were obtained from isolated and unconnected GFP-positive cells using an Axopatch 200B amplifier and Digidata 1440A controlled by pClamp10 software (Molecular Devices, Sunnyvale, CA, USA). Patch electrodes with resistance 1–3 MΩ were made from borosilicate glass (World Precision Instruments, Sarasota, FL, USA) using a linear multi-stage puller (Sutter Instruments, Novato, CA, USA) and fire polished before use. Currents were filtered at 2–5 kHz and sampled at 10 kHz. Series resistance compensation of >80% was applied to all recordings. UV-irradiation was performed simultaneously with voltage commands using a TTL controlled UVICO continuous UV light source equipped with an automated shutter (Rapp OptoElectronic, Hamburg, Germany). For single channel recordings, methods are as previously published (*Werry et al., 2013*;*Eldstrom et al., 2015*) except that the patch electrodes were fabricated with thick-walled borosilicate glass (Sutter Instruments) and a change was made to the pipette solution as detailed below.

In initial experiments, a UV-specific rundown was observed in wild type channels. We hypothesized this was due to the formation of oxidative radicals upon UV irradiation. However, supplementing the internal and external solutions with reactive oxygen species scavengers (DTT, DMSO, ascorbate, GSH) had no effect on the level of rundown (data not shown).

## Solutions

For whole-cell ionic current recordings the intracellular pipette solution contained the following (mM): 130 KCl, 5 EGTA, 1 $MgCl_2$, 4 $Na_2$-ATP, 0.1 GTP, 10 HEPES, pH adjusted to 7.2 with KOH. The extracellular bath solution contained (mM): 135 NaCl, 5 KCl, 1 $MgCl_2$, 2.8 NaAcetate, 10 HEPES, pH adjusted to 7.4 with NaOH. For single channel recordings the patch pipette solution contained (mM): 6 NaCl, 129 MES, 1 $MgCl_2$, 10 HEPES, 5 KCl, 1 $CaC1_2$ and was adjusted to pH 7.4 with NaOH. The bath solution contained (mM): 135 KCl, 1 $MgCl_2$, 1 $CaCl_2$, 10 Hepes and was adjusted to pH 7.4 with KOH

## Data analysis

G-V relations were obtained from normalized tail current amplitudes. $V_{1/2}$ and *k*-factors were obtained by fitting the data from each cell with a Boltzmann sigmoidal function. The effect of UV-crosslinking was expressed in diary plots normalized to the peak current prior to UV exposure for each cell. To account for the UV-rundown, normalized wild type currents vs. cumulative UV exposure for each cell were fit with a one-phase decay equation to determine a mean rate constant ($K_{Rundown (RD)}$: KCNQ1+KCNE1, 0.1864 $s^{-1}$; EQQQQ+KCNE1, 0.1588 $s^{-1}$; EQQ+KCNE1, 0.1049 $s^{-1}$; EQ+-KCNE1, 0.1259 $s^{-1}$). The rate of crosslinking ($K_{XL}$) was determined by plotting normalized F57Bpa containing currents vs. cumulative UV and fit with a two-phase decay equation where the slow rate

constant was set to the $K_{RD}$ obtained for each construct. Recordings where peak currents were greater than 20 nA or smaller than 2 nA or with a ratio of peak current/initial current < 5 prior to UV exposure were excluded from analysis.

## Western blot

To confirm rescue expression of F57Bpa KCNE1, tsA201 cells were transfected as described above in the presence and absence of 1 mM Bpa. After 48 hr of expression, cells were washed in PBS, lifted and pelleted by centrifugation for 1 min at 1000 x *g*. Pellets were stored at -80°C until used. Cells were lysed in 50 mM tris-HCl pH 7.4, 300 mM NaCl, 1% (w/v) DDM and protease inhibitors (Roche, Basel, Switzerland) by sonication. Lysates were clarified by centrifugation for 5 min at 2000 x *g*. Resulting supernatants were quantified and diluted into SDS-PAGE sample buffer. 3 µg of KCNE1-GFP and 50 µg of KCNE1-Y40TAG GFP or F57TAG KCNE1-Y40TAG GFP lysates were separated on a 12% bis-tris gel using MES running buffer. Proteins were then transferred to nitrocellulose for 1 hr at 100 V in NuPage transfer buffer. Blots were blocked in TBS-t containing 5% (w/v) milk powder, probed with an anti-GFP monoclonal (GF28R; Thermo Fisher Scientific) primary antibody and a goat anti-mouse HRP conjugated secondary antibody (Jackson Immunoresearch, West Grove, PA, USA) for 1 hr each at room temp. Exposures were made on film (Eastman Kodak, Rochester, NY, USA) using the Western Lightning Plus-ECL substrate (Perkin Elmer, Waltham, MA, USA).

## Co-immunoprecipitation

F57Bpa KCNE1-GFP or wild type KCNE1-GFP were co-expressed with KCNQ1 containing an N-terminal T7 affinity epitope in a tsA201 cell line stably expressing the inward rectifier $K_{ir}4.1$ (Gift from Dr. C. Ahern). Cells were washed and lifted in a low potassium hyperpolarizing solution (-90 mV; as determined by the Nernst equation) (mM): 4 KCl, 140 NMDG, 10 HEPES, 1.5 $CaCl_2$, 1 $MgCl_2$ pH 7.4. Cells were exposed to UV irradiation using a 500 W arc lamp with a Mercury-Xenon bulb (Newport Corporation, Irvine, CA, USA) for 0, 2 or 5 min. Cells were collected by centrifugation and stored at -80°C until used. Cell pellets were lysed as described above. 500 µg of each supernatant was diluted to 0.5% (w/v) DDM and 2 µg of T7-tag monoclonal (69522; Millipore, Etobicoke, ON, Canada) was added. Samples were rotated for 1 hr at room temp. 25 µl of washed, packed protein G-agarose beads (Roche) were added to each sample and rotated for 1 hr at room temp. Beads were washed with PBS containing 0.5% (w/v) DDM and 600 mM NaCl. Protein complexes were eluted with PBS containing 2% (w/v) SDS and 50 mM DTT and separated on a 7% tris-acetate gel. Western blots were performed as described above using the anti-GFP monoclonal or T7-tag monoclonal (69522; Millipore, Etobicoke, ON, Canada) antibodies.

## Statistics

Results are expressed as mean ± S.E. and represent data from at least 3 independent experiments. Difference of means testing was performed using one-way anova with Tukey's post hoc test or a two-tailed Student's t-test. A p-value of less than 0.05 was considered to be statistically significant. Non-linear regressions were performed on G-V relations using the Boltzmann sigmoidal function, and on UV-crosslinking data using the one- or two-phase decay functions in Prism 6 (GraphPad Software, La Jolla, CA, USA).

## Acknowledgements

We thank Drs. Sam Goodchild and Christopher Ahern for help with unnatural amino acid crosslinking.

## Additional information

### Funding

| Funder | Grant reference number | Author |
|---|---|---|
| Heart and Stroke Foundation of Canada | Operating Grant | David Fedida |

| Canadian Institutes of Health Research | Operating Grant | David Fedida |
| Canadian Institutes of Health Research | Postdoctoral fellowship | Christopher I Murray |

The funders had no role in study design, data collection and interpretation, or the decision to submit the work for publication.

## Author contributions

CIM, JE, Conception and design, Acquisition of data, Analysis and interpretation of data, Drafting or revising the article; MW, Acquisition of data, Analysis and interpretation of data, Drafting or revising the article; ET, RE, Acquisition of data, Analysis and interpretation of data; DF, Conception and design, Analysis and interpretation of data, Drafting or revising the article

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
