## [Decision Letter]

Thank you for submitting your work entitled "Unnatural amino acid photo-crosslinking of the *I_Ks_* channel complex demonstrates a KCNE1:KCNQ1 stoichiometry of up to 4:4" for consideration by *eLife*. Your article has been reviewed by three peer reviewers, and the evaluation has been overseen by Richard Aldrich as the Senior Editor and Reviewing Editor.

The reviewers have discussed the reviews with one another and the Reviewing editor has drafted this decision to help you prepare a revised submission.

Summary:

KCNE1 associates to KCNQ1 to form the *I_Ks_* channel in cardiac myocytes that is important for the duration of action potentials and regulation of heart rhythm. It is known that in exogenous cells different KCNE1:KCNQ1 expression ratio result in currents with various activation properties. If these various properties also exist in heart cells they will affect heart rhythm. Therefore, this phenomenon is potentially of physiological importance. The question is whether this phenomenon is due to a different KCNE1:KCNQ1 stoichiometry in the *I_Ks_* channel complex? There have been two camps with different views on this issue. One view is that the KCNQ1:KCNE1 stoichiometry varies from 4:1 to 4:4 in the *I_Ks_* channel complex, and the other view is that the *I_Ks_* channel has a fixed 4:2 KCNQ1:KCNE1 stoichiometry. Both camps have provided compelling evidence using various methods to support their view. This manuscript by Murray et al. uses two novel methods, namely single recordings and unnatural amino acid crosslinking, to argue that the *I_Ks_* channel can have different KCNQ1:KCNE1 stoichiometry and the number of KCNE1 in the channel can be up to 4. At the core of this work are two neatly characterised and well-behaving tandem constructs, that behave more or less exactly the way that we would hope. Given the various troubles and pitfalls of tandems (which seem to work better in K-channels than most), these data are encouraging, and provide a lot of weight for the concepts which the authors wish to advance. The single channel recordings are nice and support the functional distinctions between the different conditions.

However, there are some problems with the interpretation of the data, and some holes. Consequently, the major point – resolving multiple stoichiometries – is not as clearly demonstrated as the text of the paper suggests. An occasional lack of precision in the writing could also be improved to give a clearer report of what has been done.

Essential revisions:

1) The title of the paper is misleading. Crosslinking is not shown. It's enticing (and perhaps correct) to imagine that E1 is immobilising the voltage sensor of Q1 – but it's not shown here, and to do so would be considerably beyond the scope of the work. Because there is no evidence of physical crosslinking, we cannot be sure that the rates of inactivation by light are due to complexes (although the carefully-measured biophysical properties support that the expected complexes are formed in the first place). The data reported could equally be related to dissociation of the complex upon UV illumination because reversion to wild-type (Q-alone) kinetics and modulation are observed. Perhaps including a Bpa mutant that had no UV inhibition effect could provide more insight. Presumably the authors have tried to do crosslinking in biochemistry. It's very hard, but a model based on UV-dependent dissociation could explain a null result in biochemical crosslinking (and would be interesting).

2) One major problem is the UV-dependent rundown in Figure 5. The experiments with photoactive amino acids are a key quantitative point for proving intermediate stoichiometry with half the number of KCNE subunits. Thus a lot rests on these particular experiments. UV-dependent rundown as reported here has not been reported with other channels that were tried with photoactive amino acids or derivatives so far (including shaker, sodium channels, or glutamate receptors) neither is such an effect reliably reported during uncaging of calcium or other caged compounds with UV, in the range that the UVICO lamp supplies (> 320 nm). So what is shown here is unusual and unfortunate.

We don't agree with the assertion that the rundown was so small that it was easily discerned – its amplitude is 50% in the example shown. It's only 3-fold slower than the EQQ constant so a kinetic separation is also difficult.

The experiments with scavengers could have been a useful start, but provide no resolution of the problem. There are other simple experiments (many of which might already have been done) that could provide insight. Was this rundown the same for all wild-type channels (including the different tandems and combinations as tried in Figure 1–Figure 3?). Is UV dependent rundown dependent on cumulative exposure to UV (i.e. change the exposure time for each exposure), or time following UV exposure (change the frequency)? Can wild-type cells be held for long periods without exhibiting rundown? Is the rundown the same in wild-type channels no matter what state (activated or resting) is exposed? Further, is a faster UV-inhibition also observed in the activated state for the Bpa-containing mutants? If not, this could be a way to separate out the effects. The authors probably thought of all this but there's little evidence of that in the paper.

More controls are needed here to be sure that UV-effects are not simply related to what subunits were transfected (which could also, unfortunately, explain the results).

The reason of UV light induced decay of the WT *I_Ks_* is not known. This result raises the concern if the UV-induced decay of KCNQ1+ F57Bpa is specifically due to Bpa photo cross-linking. A verification of cross-linking by some other experiment may help resolve this concern.

3) Figure 2. Presents a difference in mean single channel conductance between EQ and EQQ. This difference derives from different occupancy of sub-conductance levels. It would provide more mechanistic insight if the authors use a more careful analysis of sub-conductance, the same as in their 2012 PNAS paper, to show the amplitude of each sub-level conductance and its occupancy in both EQ and EQQ channels.

4) The authors assert that EQ forms a channel with four KCNE1 and EQQ with two. While this is likely, is it possible that EQ has only two E1s associated with the channel and EQQ only has one, while other E1's from the constructs are not associated with the channel? For single channel studies, a channel with 1 or 3 KCNE1 should be included in addition to the presented data.

To be really picky, there's nothing here that says that 4 KCNE subunits are ever incorporated. If a very strong mechanism limits the incorporation to 2 E subunits, all the results here are also possible with 2:4 and 1:4 stoichiometries, for example; the lower could be obtained in EQQ due to a dose effect. The 2 vs. 4 KCNE discussion rests on the idea that 2 or 4 are incorporated if supplied by tandems. A tandem tetramer with a single KCNE subunit could also be made presumably (EQQQQ). If EQQQQ was distinct from EQQ and Q alone (or QQQQ or QQ), that would provide very strong evidence for the idea that channels containing 2 and 4 KCNE-subunits have been analysed here.

---

## [Author Response]

*Essential revisions: 1) The title of the paper is misleading. Crosslinking is not shown. It's enticing (and perhaps correct) to imagine that E1 is immobilising the voltage sensor of Q1* – *but it's not shown here, and to do so would be considerably beyond the scope of the work. Because there is no evidence of physical crosslinking, we cannot be sure that the rates of inactivation by light are due to complexes (although the carefully-measured biophysical properties support that the expected complexes are formed in the first place). The data reported could equally be related to dissociation of the complex upon UV illumination because reversion to wild-type (Q-alone) kinetics and modulation are observed. Perhaps including a Bpa mutant that had no UV inhibition effect could provide more insight. Presumably the authors have tried to do crosslinking in biochemistry. It's very hard, but a model based on UV-dependent dissociation could explain a null result in biochemical crosslinking (and would be interesting).*

Response: The reviewers raise an important oversight in our original submission. To correct this issue we have provided the readers with the biochemical data we had demonstrating the physical and specific crosslinking of F57Bpa KCNE1-GFP with KCNQ1 with UV exposure (Figure 5—figure supplement 2). We believe sharing this additional experiment eliminates the possible interpretation that KCNE1 is dissociating from KCNQ1 upon UV exposure. See subsection “UV-dependent channel crosslinking”, second paragraph.

*2) One major problem is the UV-dependent rundown in Figure 5. The experiments with photoactive amino acids are a key quantitative point for proving intermediate stoichiometry with half the number of KCNE subunits. Thus a lot rests on these particular experiments. UV-dependent rundown as reported here has not been reported with other channels that were tried with photoactive amino acids or derivatives so far (including shaker, sodium channels, or glutamate receptors) neither is such an effect reliably reported during uncaging of calcium or other caged compounds with UV, in the range that the UVICO lamp supplies (> 320 nm). So what is shown here is unusual and unfortunate. We don't agree with the assertion that the rundown was so small that it was easily discerned* –

*its amplitude is 50% in the example shown. It's only 3-fold slower than the EQQ constant so a kinetic separation is also difficult. The experiments with scavengers could have been a useful start, but provide no resolution of the problem. There are other simple experiments (many of which might already have been done) that could provide insight. Was this rundown the same for all wild-type channels (including the different tandems and combinations as tried in Figure 1–Figure 3?). Is UV dependent rundown dependent on cumulative exposure to UV (i.e. change the exposure time for each exposure), or time following UV exposure (change the frequency)? Can wild-type cells be held for long periods without exhibiting rundown? Is the rundown the same in wild-type channels no matter what state (activated or resting) is exposed? Further, is a faster UV-inhibition also observed in the activated state for the Bpa-containing mutants? If not, this could be a way to separate out the effects. The authors probably thought of all this but there's little evidence of that in the paper. More controls are needed here to be sure that UV-effects are not simply related to what subunits were transfected (which could also, unfortunately, explain the results). The reason of UV light induced decay of the WT I_Ks_ is not known. This result raises the concern if the UV-induced decay of KCNQ1+ F57Bpa is specifically due to Bpa photo cross-linking. A verification of cross-linking by some other experiment may help resolve this concern.*

Response: To address the reviewers’ concerns regarding the nature of the UV-rundown we observed in the crosslinking experiments we have redone some of our analysis, added new data to Figure 5, added two additional supplementary figures and amended the text.

In Figure 5 the UV effect for both the F57Bpa and wild type KCNE1 constructs are presented (diary plots in panels B, C D and E) to demonstrate that the UV rundown observed was equivalent across all channel configurations. We included difference of means testing to compare the normalized peak currents of the Bpa and wild type channels after 20 sweeps of UV (panel B, C, D and E). We also compared normalized peak currents across all channel configurations after 2.1 s of UV exposure. Finally, we have reanalyzed the crosslinking rate data, including the rundown data obtained for EQQQQ, EQQ and EQ co-expressed with wild type KCNE1, to account for any differences (See [Supplementary-material SD4-data]). The UV-rundown analysis for all the constructs is presented in Figure 5—figure supplement 1. The changes to our analysis are detailed in the Methods section about data analysis. Using this more rigorous experimental design and analysis the EQQ crosslinking rate is now more than 6-fold greater than the background rate observed for EQ.

Additionally, as discussed in our response to Essential Revision 1, we have provided an alternate method for determining a covalent crosslink between KCNE1 and KCNQ1 in Figure 5—figure supplement 2.

For the benefit of the reviewers, we have also provided an additional figure comparing the rundown observed in wild type *I_Ks_* and K_v_1.5 channels with and without UV-irradiation. As anticipated by the comments, the same UV-rundown was not observed in K_v_1.5 channels. K_v_1.5 channels exhibited approximately a 20% loss in peak current over 20 sweeps in the presence or absence of UV exposure (See Figure 7). We observed a similar ~20% rundown for wild type *I_Ks_* alone but a significantly larger rundown (~50% of the original current) with UV indicating that the *I_Ks_* channel is susceptible to a non-specific UV effect.

Author response image 1.UV-rundown is not present in K_v_1.5 channels.Schematic of the voltage protocols in the absence (**A**) and presence (**B**) of UV application. Below are representative K_v_1.5 currents showing rundown response with and without UV exposure. Current trace for KCNQ1 + wt KCNE1 showing rundown in the absence (**C**) and presence (**D**) of UV. (**E**) Combined diary plot is shown of normalized peak currents for K_v_1.5 – UV (grey diamonds), +UV (blue diamonds), and KCNQ1 + wt KCNE1 – UV (black circles), +UV (red circles; data from Figure 5). (**F**) Comparison of normalized peak currents after 2.4 s of cumulative UV exposure (indicated by arrow in panel D) (n = 3-5; * = p<0.001).**DOI:**
http://dx.doi.org/10.7554/eLife.11815.020

While we agree with the reviewers that this is an unfortunate reality of our experimental system, we believe we have sufficiently characterized this effect to conclude that we are measuring a Bpa-specific UV crosslinking effect and our findings can be resolved from the background effects.

*3) Figure2. Presents a difference in mean single channel conductance between EQ and EQQ. This difference derives from different occupancy of sub-conductance levels. It would provide more mechanistic insight if the authors use a more careful analysis of sub-conductance, the same as in their 2012 PNAS paper, to show the amplitude of each sub-level conductance and its occupancy in both EQ and EQQ channels.*

Response: We have performed the requested analysis and presented the results in Figure 2—figure supplement 2. These data are discussed in the second paragraph of the subsection “Reduced single-channel conductance and latencies of EQQ and EQQQQ *I_Ks_* channel complexes”.

4) The authors assert that EQ forms a channel with four KCNE1 and EQQ with two. While this is likely, is it possible that EQ has only two E1s associated with the channel and EQQ only has one, while other E1's from the constructs are not associated with the channel? For single channel studies, a channel with 1 or 3 KCNE1 should be included in addition to the presented data. To be really picky, there's nothing here that says that 4 KCNE subunits are ever incorporated. If a very strong mechanism limits the incorporation to 2 E subunits, all the results here are also possible with 2:4 and 1:4 stoichiometries, for example; the lower could be obtained in EQQ due to a dose effect. The 2 vs. 4 KCNE discussion rests on the idea that 2 or 4 are incorporated if supplied by tandems. A tandem tetramer with a single KCNE subunit could also be made presumably (EQQQQ). If EQQQQ was distinct from EQQ and Q alone (or QQQQ or QQ), that would provide very strong evidence for the idea that channels containing 2 and 4 KCNE-subunits have been analysed here.

Response: To further differentiate the 4:4 vs. 2:4 stoichiometry we have generated and evaluated an EQQQQ construct. We have added data involving this additional construct to all the relevant figures (Figure 1, Figure 2, Figure 3 and Figure 5). In performing these experiments we were very pleased to observe that EQQQQ had a further hyperpolarized V0.5, shorter first latency, and reduced peak conductance compared to EQQ, all of which were reversed with co-expression of KCNE1. EQQQQ also had intermediate rate of crosslinking compared to Q1 and EQQ, consistent with the association of three KCNE1 subunits in the three available clefts. Incorporation of the new data required considerable changes to several figures and expansion of the manuscript.